# Terahertz Rectangular Waveguides by UV-LIGA with Megasonic Agitation

**DOI:** 10.3390/mi13101601

**Published:** 2022-09-26

**Authors:** Yongtao Li, Yi Wang, Hanyan Li, Bai Jiang, Fan Bai, Jinjun Feng

**Affiliations:** 1School of Mechanical and Automotive Engineering, Guangxi University of Science and Technology, Liuzhou 545006, China; 2School of Automation, Guangxi University of Science and Technology, Liuzhou 545006, China; 3Beijing Vacuum Electronics Research Institute, Beijing 100015, China

**Keywords:** terahertz, UV LIGA, megasonic agitation, rectangular waveguide, WR2.8

## Abstract

This paper researches the fabrication of a WR2.8 terahertz rectangular waveguide operating at the frequency ranging from 260 GHz to 400 GHz via UV-LIGA technology (UV-lithography, electroplating, and molding). In the process, megasonic agitation is applied to improve the mechanical properties and internal surface roughness of the WR2.8 rectangular waveguide. The effects of process parameters on the properties of structures are discussed, and optimized parameters are obtained to achieve accurate geometry dimensions. In addition, the highly crosslinked SU-8 is reliably removed from structures without damage through a synthesis method. The accuracy of the height and width of the WR2.8 rectangular waveguide is 5 µm and 2 µm, respectively, and the measured internal surface roughness is 79.6 nm. Moreover, experimental measurements and numerical simulations of the waveguide are conducted, and the difference between the two is also highlighted.

## 1. Introduction

The terahertz spectrum spans the millimeter to submillimeter wave bands (from 0.1 to 10 THz) [1], and it has great application prospects in physics, materials science, communications, astronomy, and radar systems [2,3,4,5]. In the process of THz technological development, several approaches are used to generate THz range radiation, such as solid-state oscillator, optical THz radiation, and vacuum electron devices [1]. However, until now, only vacuum electronic devices have been able to generate high-power THz radiation to satisfy some special applications. To integrate THz vacuum electronic systems, the most fundamental THz component is the rectangular metallic waveguide due to its advantages of high power capacity and low transmission loss.

As the frequency goes up to the terahertz band, the typical size of a rectangular waveguide gradually decreases to the submillimeter scale, and the size tolerance is required to be in the micron scale. In addition to high machining accuracy, terahertz waveguides also have strict requirements on surface roughness, mainly because the skin depth of the terahertz electromagnetic wave becomes smaller, reaching the nanometer scale. It can be seen that with the increase in frequency, the dimensions of the rectangular waveguide decrease, and the requirement of surface roughness increases, so advanced micromachining technology is employed to meet the requirements [6,7].

At present, the advanced micromachining technology that can fabricate terahertz rectangular waveguides mainly includes high-speed milling, precision electrical discharge machining (EDM), 3D printing, the laser micromachining technique, UV-LIGA [8,9,10] and deep reactive ion etching (DRIE) [11,12,13] Sumer Makhlouf fabricated a WR3 rectangular waveguide (operating at 230–320 GHz) using metallic laser beam melting (LBM) 3D printing, with the average surface roughness of 2 µm and THz insertion loss of 0.3 dB/mm [14]. Gagan Kumar designed, fabricated and characterized a planar THz waveguide technology based on a 1D array of periodically spaced blind holes made using standard laser micromachining techniques [15]. Shashank Pandey demonstrated a slot-waveguide-based splitter for broadband terahertz radiation using a T-shaped waveguide structure [16]. William R used the silicon micromachining technique to create silicon-based WR10 waveguides (operating at 75–110 GHz), with surface roughness of 10 µm and insertion loss of 0.04 dB per wavelength (at 100 GHz) across most of the band [17].

In recent years, we have made an effort to microfabricate terahertz components using UV-LIGA technology, such as terahertz rectangular waveguides, filters and splitters with dimensions of 1–1000 μm. This paper provides a new idea for the development and adaption of UV-LIGA technology to produce rectangular waveguides with frequencies up to terahertz. The UV-LIGA technology process and parameter optimization for WR2.8 rectangular waveguides are described in Section 4, in addition to comparisons of various SU-8 removal methods. In Section 5, S parameters of the WR2.8 rectangular waveguide are tested and simulated. We also highlight the differences between the two.

## 2. Designing Variables of the Terahertz Waveguide

Figure 1a,b show the design of the WR2.8 rectangular waveguide structure and its parameters. It operates at the frequency ranging from 260 GHz to 400 GHz. The WR2.8 rectangular waveguide is composed of three parts: a waveguide half with a rectangular channel (hereafter called a channel block), a copper plate cover and standard UG-387 flanges, as shown in Figure 1b. The sectional view of the WR2.8 waveguide is shown in Figure 1a, with the width of a = 710 μm, height of b = 355 μm and length of 25.4 mm. According to the dimension and feature of each part, UV-LIGA technology is proposed to fabricate the channel block, the copper plate cover is cut via WEDM (Wire Electrical Discharge Machining), and the flanges are shaped using the traditional machining method. Finally, three parts are assembled to form the WR2.8 rectangular waveguide. The waveguide is fully enclosed, leaving the input and output ports. This approach is chosen to simplify the microfabrication process and improve the electromagnetic performance.

## 3. Fabrication Process

In the UV-LIGA process to fabricate the channel block, the wafers used are oxygen-free copper discs with 8 mm thicknesses and 3-inch diameters. Oxygen-free copper is used in the experiment due to its good properties of electrical and thermal conductivity. A lathe is used to flatten the surfaces of the oxygen-free copper discs. Then, the wafers are grinded and polished on a single side, with surface roughness of about 10 nm, followed by strict cleaning. Cleaning is performed to ensure no contaminants remain on the surface of the wafer.

The process of UV-LIGA technology is shown in Figure 2, which is as follows: (1) A layer of photoresist is spin-coated on the wafer following by a soft bake. (2) The wafer is exposed to UV light. (3) The resist is developed in the developer. (4) Copper is deposited onto the wafer via electroformation. (5) The surface of the copper-coated wafer is grinded and polished. (6) The photoresist is removed. (7) The sample is separated to obtain individual channel blocks. (8) The channel block is bonded with a copper plate cover.

Finally, the complete electromagnetic transmission channel of the WR2.8 waveguide is obtained, which is shown in Figure 3.

## 4. Improvement Methods and Discussion

SU-8 2150 (Microchem, Austin, TX, USA) is used in the fabrication process due to its advantages of good mechanical properties, chemical corrosion resistance capacity and excellent physical properties. Its viscosity is 80,000 cSt, and it can achieve a broad range of thickness via spin-coating. In this experiment, 400 μm-thick SU-8 resist film was coated onto an oxygen-free copper disc for the lithography process.

### 4.1. Fabricating the SU-8 Resist Film

To achieve the 400 μm-thick SU-8 resist film, we poured 3 mL of SU-8 resist onto the wafer; next, we spun the wafer at 100 rpm for 20 s with acceleration of 50 rpm/s, and then it was spun at 400 rpm for 40 s with acceleration of 100 rpm/s. The resist thickness is mainly determined by the spin speed. According to experimental data, the curve of spin speed versus thickness was achieved, which is shown in Figure 4, which was different from the one provided by Microchem.

Then, the wafer was placed on a level hotplate at 50 °C and held for 2 h, which had been placed horizontally with a level meter in advance. The next step in the process was a soft bake, starting at 65 °C and ramping up to the temperature of 105 °C. If the temperature exceeds 130 °C, SU-8 resist will be prematurely cross-linked and will be undeveloped, which is called heat cross-linking, as shown in Figure 5. In this experiment, the SU-8 resist was soft-baked at 65 °C for 30 min and 105 °C for 4 h on the level hotplate, and then cooled down to room temperature. Table 1 shows the soft bake temperatures and time for the SU-8 2150 resist layer with a thickness of 400 μm. After the soft bake, the thickness of the photoresist layer was 400 μm, and the uniformity was ± 5%.

### 4.2. Exposure

SU-8 resist was exposed with i-line UV light on a contact mask aligner, which was loaded with an i-line filter. During the process, the PAG contained in SU-8 resist created photo-acid in the exposed areas, which acted as a catalyst in the subsequent cross-linking reaction occurring in the post-bake process.

In order to achieve the proper exposure dosage, we present some results from four representative exposure experiments, as shown in Figure 6. First, the patterns were fabricated at the relatively high dose of 1500 mJ/cm^2^. Figure 6a shows an optical microscope photograph of the SU-8 pattern with the top linewidth of 720 μm, wider than the mask linewidth of 710 μm. A high exposure dosage created excessive photo-acid, diffusing into the unexposed region. The excessive photo-acid caused cross-linking where no cross-linking should occur, resulting in a wider linewidth. As shown in Figure 6b, the top linewidth was 714 μm. It is obvious that.

1300 mJ/cm^2^ was still a little high. The third resist pattern was obtained under a proper exposure dosage of 1100 mJ/cm^2^, as shown in Figure 6c. From Figure 6c, the top linewidth was 710 μm, the same as the mask linewidth. Contrary to the cases in Figure 6a,b,d shows the top linewidth was 705 μm at the exposure dosage of 900 mJ/cm^2^, narrower than the mask linewidth. SU-8 resist was underexposed and unable to create enough photo-acid to cross-link SU-8 molecules. This case might have permitted the developer to diffuse into the boundary of the underexposed region, causing the linewidth to become narrower. Table 2 shows the optimum exposure dose for the 400 μm-thickness SU-8 layer.

### 4.3. Post-Exposure Baking and Developing

Post-exposure baking was carried out to cross-link SU-8 molecules. The SU-8 wafer was baked on a hotplate, being heated up to 65 °C at 2 °C min, held at 65 °C for 15 min, then ramped up to 90 °C, and held at 90 °C for 20 min. During the post-exposure bake, it is undesirable to over-bake the SU-8 resist. If the resist is over-baked, excessive stress will be produced in the exposed region due to extra cross-linking, causing cracks in the SU-8 patterns. On the contrary, if there is under-baking post-exposure, SU-8 molecules will not fully cross-link, and the SU-8 pattern will be soft, just as it would be if it was under-exposed.

After post-exposure baking, the resist was developed in MicroChem’s SU-8 developer. The SU-8 developer dissolved the unexposed area, leaving the SU-8 patterns. In the experiment, the sample was put into the beaker filled with developer. The sample could be fully developed after 15 min of development. Table 3 shows the post-exposure bake temperatures and time and the development time for the resist layer. Next, the sample was rinsed with IPA, then washed with deionized water, and dried with an air stream. Figure 7 shows the pattern after the development. From Figure 7, it can be seen that there was no residual photoresist on the surface of the sample, which provided a good electroforming mold for the subsequent electroforming process.

### 4.4. Copper Electroforming

Copper electroforming deposits copper onto surfaces between the cross-linked SU-8 patterns. The mechanism of copper electroforming is based on two electrochemical reactions taking place in electrodes in an electrochemical cell containing an electrolyte. The copper-electroforming electrolyte is composed of a CuSO_4_•5H_2_O and H_2_SO_4_ solution with additives. In the experiment, megasonic agitation was applied to improve the quality of the copper layer due to its cavitation phenomena such as acoustic streaming, shock waves, and surface cleaning. The precision electroforming equipment used consisted of the electrochemical cell, a electroforming power supply, a temperature control unit, a filtering circulation system, and megasonic transducers wherein the copper balls acted as the anode, and the sample with SU-8 patterns to be electroformed acted as the cathode. Prior to electroforming, the sample was etched using plasma to remove contaminants. The current density, defined as the ratio of the current to the area of black surface of the sample, was about 10 mA/cm^2^, yielding a deposition rate of approximately 10 μm/h. The power of mega sound introduced into the deposition process was 200 W at a frequency of 1 MHz. Figure 8 shows the raw copper surface of the sample after electroformation for 40 h. From the picture, there are many knots on the copper surface, perhaps because additives were consumed too much or were lower than the optimal level. Table 4 shows the copper electroforming parameters in the process.

### 4.5. SU-8 Removal

The electroformed sample was soaked in MicroChem Remover PG at a temperature of 70 °C for several tens of hours, assisted by ultrasound. The SU-8 resist softened, swelled, and was partially removed. Then, the sample was placed in a hydrogen furnace at 600 °C for 15 min. After being cooled down to room temperature, the sample was taken out, and it was found that the resist left in the channels had burned out into carbonized solids, which could be cleared away with an acetone solution in an ultrasonic cleaner for 15 min. The surfaces at the bottom of the channels after treatment were smooth and were not damaged in the process. The channel was 710 μm wide. Clearly, the synthesis method is the most effective method to remove the fully cross-linked SU-8 resist compared to the abovementioned methods. Figure 8 shows the sample after SU-8 removal using the synthesis method. It is clear that no resist was left in the channels. The sample had many knots and turned black, partly due to the carbonized resist in the hydrogen furnace, which was treated in the subsequent cutting. 

### 4.6. Cutting and Assembling

With WEDM (Wire Electrical Discharge Machining), the sample was cut into individual copper channel blocks, as shown in Figure 9. It can be seen from Figure 9b that the cross-section of the channel was rectangular, which was 710 ± 1 µm wide and 355 ± 2 µm high. The copper channel blocks and the cover were assembled and put into a special clamp, which held them together tightly. Then, they were placed into a vacuum furnace, heated up to 850 °C, and held for 1 h.

After being cooled down to room temperature, the complete electromagnetic transmission channel of the WR2.8 waveguide was formed, as shown in Figure 10. According to Figure 10b, the dimension of the cross-section was 710 ± 1 µm wide and 357 ± 2 µm high, which was not deformed in the process of bonding. Table 5 shows a comparison between the microfabricated and designed dimensions.

The complete electromagnetic transmission channel of the WR2.8 waveguide was machined again according to the flange sizes, as shown in Figure 11. Then, the structures were assembled with the flanges, forming the WR2.8 waveguide.

## 5. Results and Analysis

The measurement of the rectangular channel of the WR2.8 waveguide shows accuracies of ±1 µm and ±2 µm for the width and height, respectively, using a microscope. Accuracy is defined as the deviation of a measurement compared with its designed value. We also measured several spots and demonstrate typical surface roughness at two different spots on the sidewall. As described in Figure 12, the sidewall roughness was 79.6 nm and 64.9 nm, respectively. Compared with the KMPR-based UV-LIGA technology in references [9,10], the sidewall is relatively rougher than that in the SU-8-based UV-LIGA technology, but the process stability and repeatability is better, especially for structures with thicknesses up to 400 μm. Thus, for structures which are produced in large quantities, such as THz metal rectangular waveguides, the SU-8 based UV-LIGA technology is more applicable.

At microwave frequencies, the induced current flows in the exceedingly thin skin of the waveguide. The skin depth of the metal is calculated using the following formulas:(1)$$\delta =\sqrt{\frac{2}{\omega \mu \sigma }}$$
where ω is the angular frequency and μ and σ are the permeability and the smooth-surface conductivity of the metal, respectively. For copper, at the frequency of 340 GHz, the calculated skin depth δ is about 1.13 × 10^−5^ cm or 113 nm. Heat losses caused by eddy currents in the thin skin will be increased due to the roughness of the waveguide surface when the roughness is comparable to the skin depth [18]. The relationship between surface roughness and effective conductivity can be written as [19].
(2)$$\sigma_{f}=\frac{\sigma }{{\left\{1+\frac{2}{\pi }arctan\left[1.4{\left(\frac{R_{a}}{\delta }\right)}^{2}\right]\right\}}^{2}}$$
where $R_{a}$ is surface roughness and $\sigma_{f}$ is effective conductivity. Figure 13 shows effective conductivity as a function of surface roughness at 340 GHz. According to Figure 13, the effective conductivity rapidly decreases between the surface roughness of 50 nm and 100 nm. As surface roughness exceeds 150 nm, its effect on the effective conductivity begins to diminish. For the WR2.8 rectangular waveguide with the surface roughness of 79.6 nm, the effective conductivity is 2.95 × 10^7^ S/m at 340 GHz, which will affect the transmission property.

In the paper, CST was used to calculate S parameters exhibiting the effect of surface roughness. The model of the WR2.8 waveguide with the size of 710 μm × 355 μm used in the CST calculation is shown in Figure 14. The metal conductivity of copper was set as 2.95 × 10^7^ S/m, corresponding to the roughness of 79.6 nm.

The simulated S_21_ for the TE10 mode was about −0.58 dB over the band. We also measured the S_21_ parameter using the two-port vector network analyzer. The flanges of the WR2.8 waveguide were used to connect flanges on the test fixture and fixed by screws. The experimentally measured S_21_ was from −2.3 dB to −2.9 dB. As shown in Figure 15, by comparing the measured and simulated results, the measured S_21_ was −2 dB less than the simulated one, caused by test errors, and mainly produced by the small dimensional tolerance and mechanical alignment. Figure 16 shows the S_11_ parameters of the waveguide. From the figure, it can be seen that the measured S_11_ parameters ranged from −16 dB to −48 dB, while the calculated S_11_ parameter was about −85 dB. Further work should be carried out to improve the assembly accuracy of waveguides and flanges and the precision of flanges, reducing electrical reflection from a mated pair of flanges.

## 6. Conclusions

We have demonstrated a new approach for the microfabrication of terahertz rectangular waveguides using UV-LIGA technology with megasonic agitation. In particular, we have microfabricated a WR2.8 copper rectangular waveguide with cross-sectional dimensions of a = 710 ± 1 µm and b = 355 ± 2 µm and the sidewall roughness of 79.6 nm. The measured result exhibited insertion loss from −2.3 dB to −2.9 dB over the entire frequency band ranging from 325 GHz to 375 GHz, less than the simulated one, mainly produced by the small dimensional tolerance and mechanical alignment. Further work should be carried out to improve assembly accuracy of waveguides and flanges. 

## Figures and Tables

**Figure 1 micromachines-13-01601-f001:**
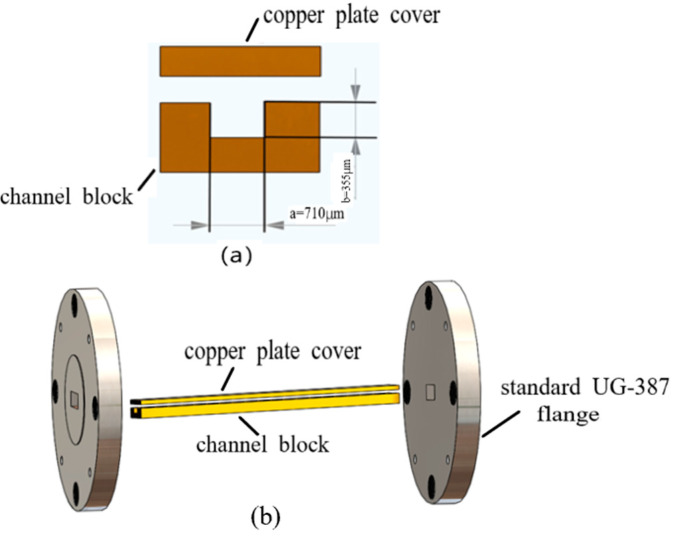
(**a**) A cross-sectional dimension of a WR2.8 rectangular waveguide: a = 710 μm and b = 355 μm. (**b**) A WR2.8 waveguide is composed of three parts: a channel block, a copper plate cover and standard UG-387 flanges.

**Figure 2 micromachines-13-01601-f002:**
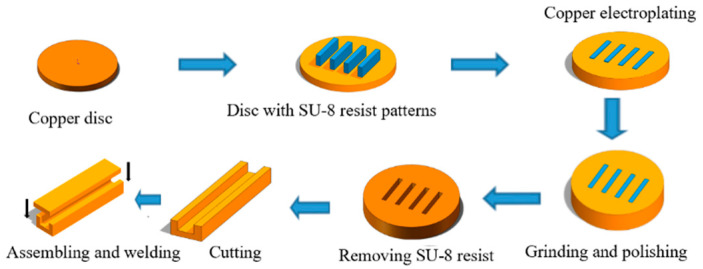
UV-LIGA process for creating the complete electromagnetic transmission channel of the WR2.8 waveguide: (1) preparing oxygen-free copper discs with 8 mm thickness and 3-inch diameter; (2) forming a SU-8 pattern using the lithography process; (4) electroforming copper between the cross-linked SU-8 patterns; (5) grinding and polishing the electroformed copper wafer to 355 μm; (6) removing SU-8 photoresist; (7) cutting the sample; (8) bonding the channel block and the copper plate cover via diffusion welding.

**Figure 3 micromachines-13-01601-f003:**
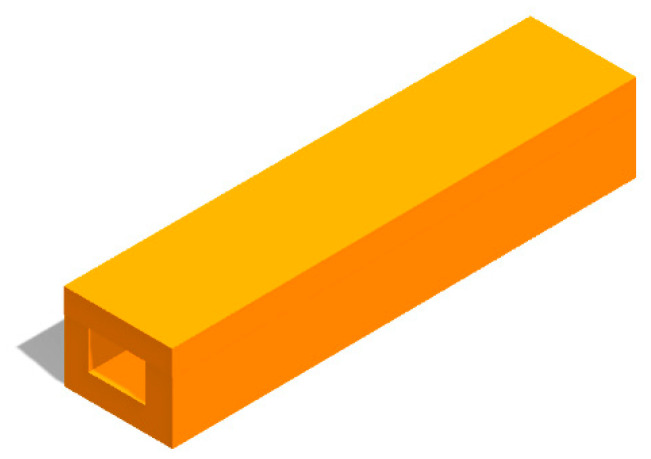
A complete electromagnetic transmission channel of a WR2.8 waveguide, consisting of a channel block and a copper plate cover.

**Figure 4 micromachines-13-01601-f004:**
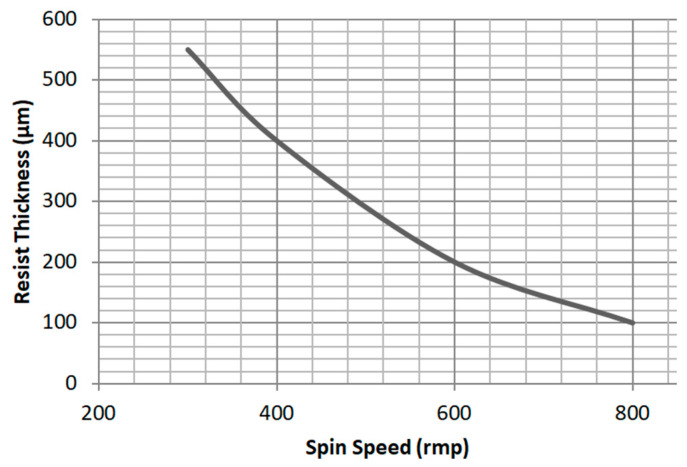
SU-8 2150 spin speed versus thickness. A spin speed of 400 rmp generates a 400 μm-thick SU-8 resist film.

**Figure 5 micromachines-13-01601-f005:**
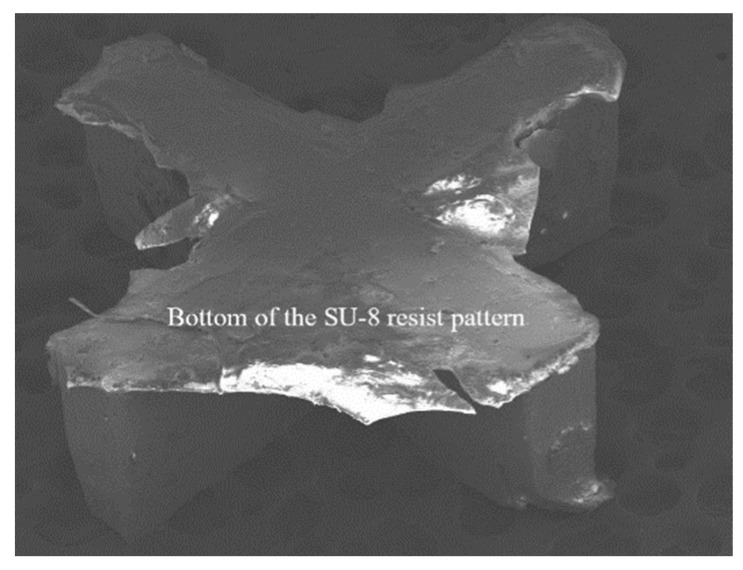
A thin resist layer on the bottom of the SU-8 resist pattern undeveloped in the developer solution, because it cross-linked in the soft bake at temperature exceeding 130 °C.

**Figure 6 micromachines-13-01601-f006:**
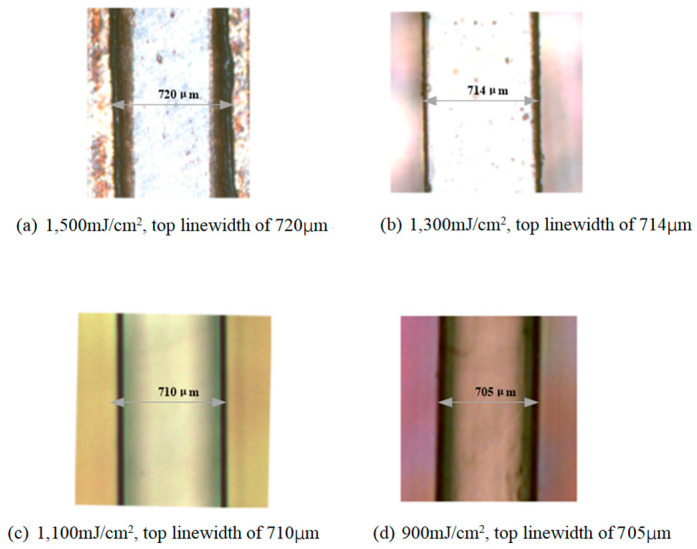
SU-8 resist patterns obtained by various exposure dosages and corresponding top linewidth.

**Figure 7 micromachines-13-01601-f007:**
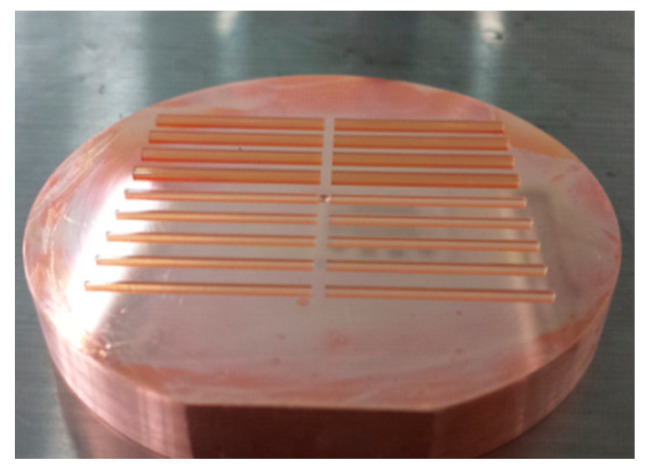
Optical microscope photograph of 18 SU-8 molds with optimum parameters on a copper disc: 8 SU-8 molds above for WR2.8 and 10 molds below for WR1.5.

**Figure 8 micromachines-13-01601-f008:**
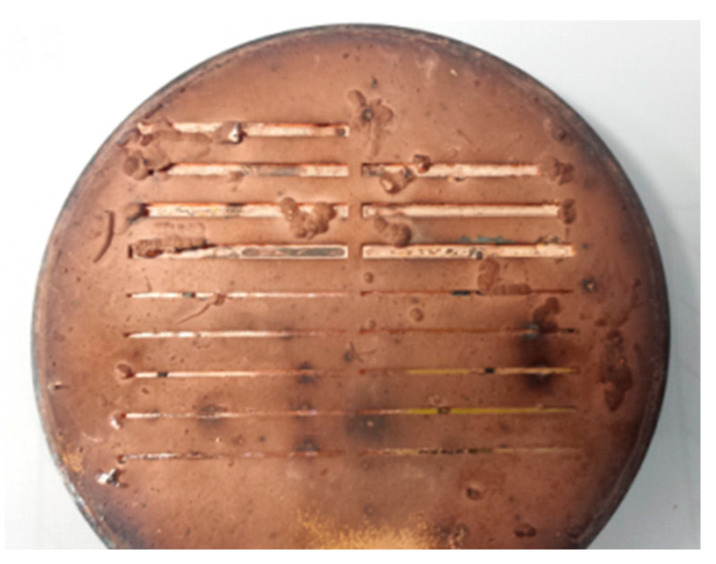
Optical microscope photograph of of the sample with 18 channels after SU-8 removal: 8 channels above for WR2.8, and 10 channels below for WR1.5.

**Figure 9 micromachines-13-01601-f009:**
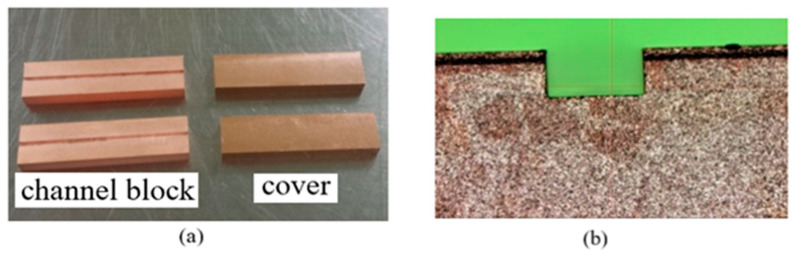
(**a**) Copper channel blocks and corresponding covers. (**b**) Cross-section of the channel.

**Figure 10 micromachines-13-01601-f010:**
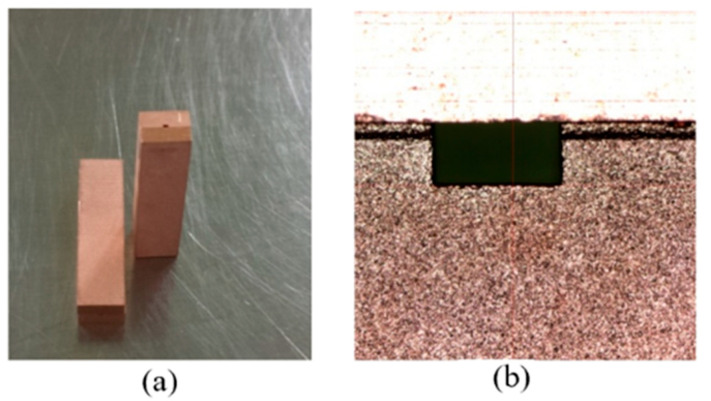
(**a**) Complete copper terahertz straight waveguide structures consisting of a channel block and a cover after bonding. (**b**) Cross-section of the waveguide structure.

**Figure 11 micromachines-13-01601-f011:**
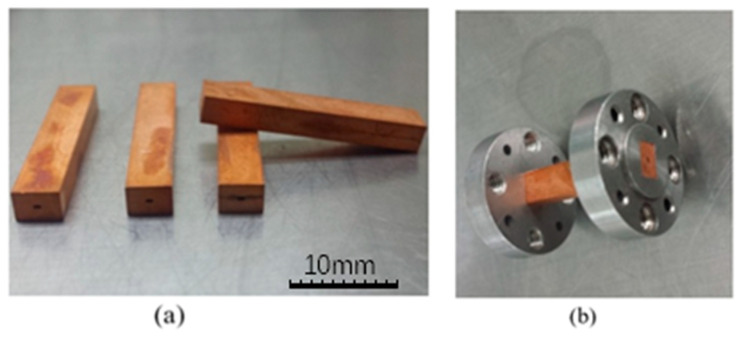
(**a**) The complete electromagnetic transmission channels of the WR2.8 waveguide after secondary machining. (**b**) The WR2.8 waveguide with flanges.

**Figure 12 micromachines-13-01601-f012:**
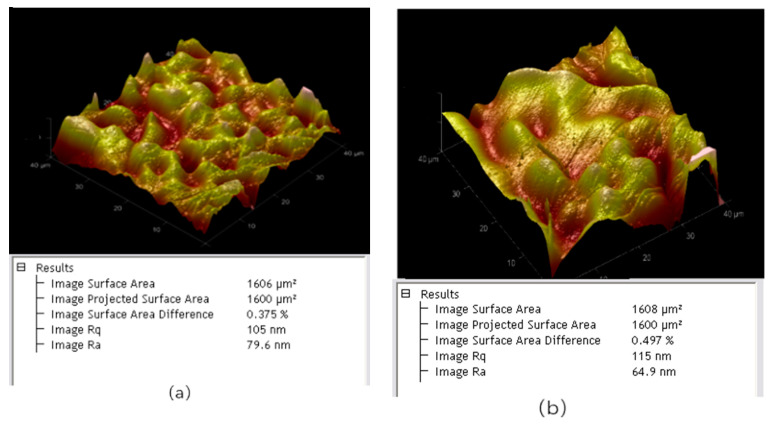
Testing sidewall roughness of the channel at two different spots with atomic force microscopy. (**a**) The sidewall roughness is 79.6 nm at one spot (Ra = 79.6 nm). (**b**) The sidewall roughness is 64.9 nm at another spot (Ra = 64.9 nm).

**Figure 13 micromachines-13-01601-f013:**
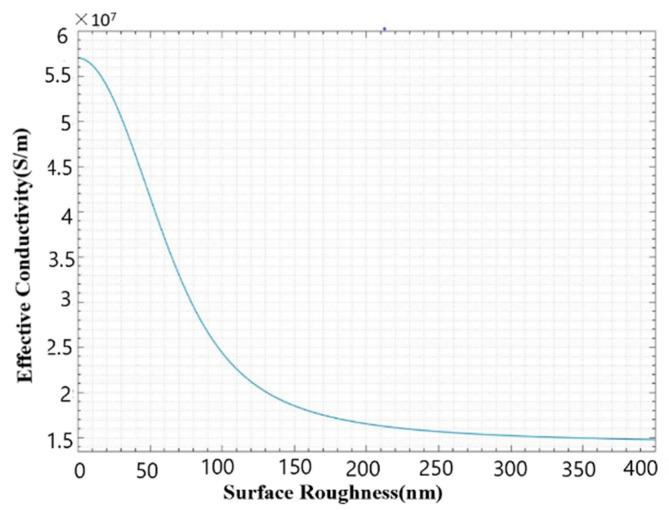
The effect of surface roughness on effective conductivity values at 340 GHz, showing the relative decrease in conductivity as a function of surface roughness.

**Figure 14 micromachines-13-01601-f014:**
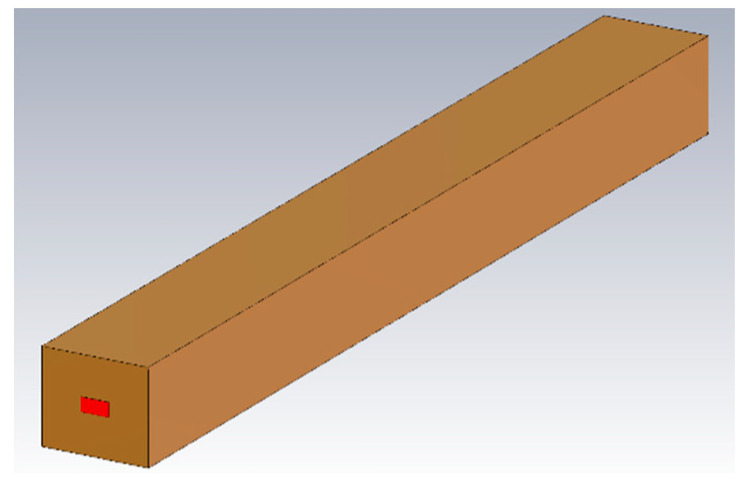
The model of the WR2.8 waveguide used in the CST calculation. The copper conductivity was set as 2.95 × 10^7^ S/m in calculation.

**Figure 15 micromachines-13-01601-f015:**
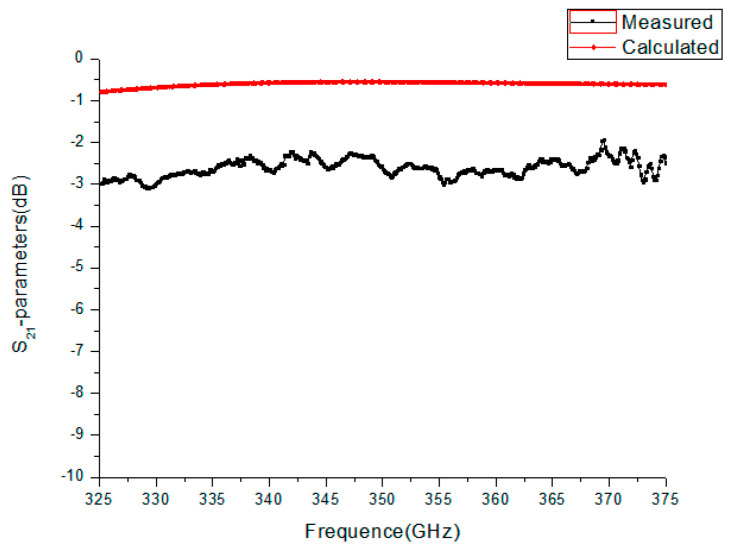
The experimentally measured S_21_ and calculated S_21_ parameters of the WR2.8 waveguide.

**Figure 16 micromachines-13-01601-f016:**
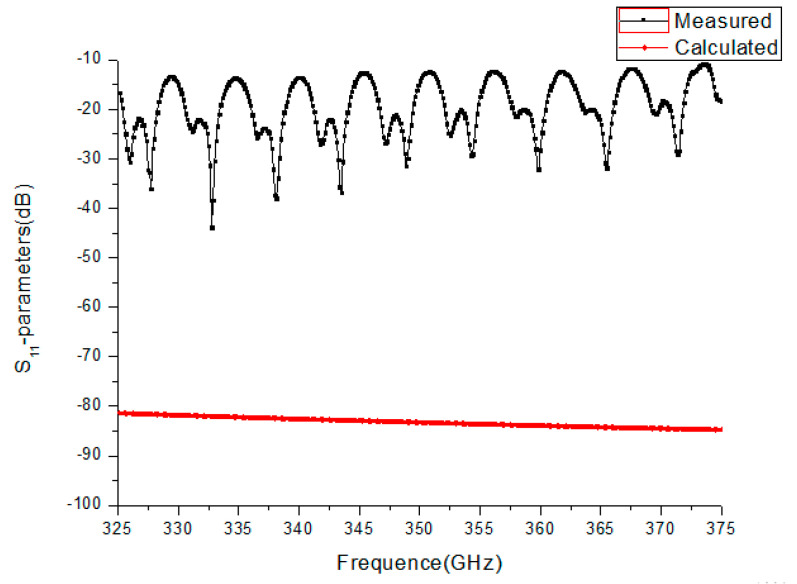
The experimentally measured S_11_ and calculated S_11_ parameters of the WR2.8 waveguide.

**Table 1 micromachines-13-01601-t001:** Soft bake temperature and time.

Thickness	Soft Bake Temperature and Time	
400 μm	65 °C	105 °C
	30 min	4 h

**Table 2 micromachines-13-01601-t002:** Optimum exposure for the SU-8 layer.

Thickness	Optimum Exposure	Linewidth
400 μm	1100 mJ/cm^2^	710 μm

**Table 3 micromachines-13-01601-t003:** Post-exposure baking parameters and development time.

Soft Baking Temperature and Time		Development Time
65 °C	90 °C	15 min
15 min	20 min	

**Table 4 micromachines-13-01601-t004:** Copper electroforming parameters.

Power of Mega Sound	Current Density	Deposition Rate	Deposition Time
200 W	10 mA/cm^2^	10 μm/h	40 h
1 MHz			

**Table 5 micromachines-13-01601-t005:** Comparison between microfabricated and designed dimensions.

Parameters	Designed	Fabrication (before Bonding)	Fabrication (after Bonding)
Width (a)	710 μm	710 ± 1 μm	710 ± 1 μm
Height (b)	355 μm	355 ± 2 μm	355 ± 2 μm

## Data Availability

The data that support the findings of this study are available from the corresponding author upon reasonable request.

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
