# Peer review of "Terahertz Rectangular Waveguides by UV-LIGA with Megasonic Agitation"

_micromachines, 2022, doi:10.3390/mi13101601_

Round 1

Reviewer 1 Report

In this work, the authors fabricated a THz metal rectangular waveguide (2.8WR rectangular waveguide). Waveguides of this type for the THz range have been known for more than 20 years (for example, 10.1364/josab.17.000851) and are widely used to extract radiation in vacuum THz sources (10.1109/tthz.2013.2255914). 

The work uses photolithography electroforming (UV-LIGA technology) to minimize the roughness of the inner walls of the waveguide channel, it is stated that the measured sidewal roughness is 79.6nm. However, earlier papers were published in which metal rectangular waveguides and other THz elements for vacuum electronics were also fabricated using a similar method:

10.1109/led.2013.2241389,  10.1109/tmtt.2018.2827383, 10.1109/IVELEC.2010.5503542, 10.1109/ted.2017.2683399.

The surface roughness of such elements is similar or even less (10.1109/led.2013.2241389) than suggested in this paper. Therefore, the scientific novelty of the work is not obvious.

At the same time, there are a number of questions about the actual content of the article:

1) The transmission channel was covered with a copper plate and connected by diffusion welding, however, the effect of defects at the points of contact on the effective conductivity and losses is not considered at all. Possible plasmon effects will worsen the structure of the field.

2) 2) It would be much more informative to give the attenuation coefficient for the waveguide, the S-parameter does not allow you to separate the input and output losses from the radiation losses directly in the waveguide (for example, in 10.1109/tmtt.2018.2827383).

3) It is impossible to compare the dependences of the S21 parameter for the calculations and the experiment (Figures 15 and 16 are screenshots of the programs, the range of axes is chosen so that all the features of the dependences are smoothed).

4) The authors measured the roughness using atomic force microscopy. For greater accuracy, it would be better to take measurements in several areas of the waveguide to make sure that the roughness is approximately the same throughout.

Thus, I do not recommend this work for publication.

Author Response

Dear Reviewers,

        We have studied your comments carefully and made a careful revision on the original manuscript. We would like to submit a revised manuscript which has revised according to the your and reviewers’ comments, a detailed point-by-point reply to all of the reviewers' comments, and a marked up manuscript where we highlight in red the changes made with respect to the initial submission. We hope you can reconsider our revised manuscript for potential publication in Micromachines.

All authors have read and approved this version of the article, and due care has been taken to ensure the integrity of the work. Neither the entire paper nor any part of its content has been published or has been accepted elsewhere. It is not being submitted to any other journal.

Thank you very much for providing a complete guide to the manuscript revision and submission. We are looking forward to hearing from you soon.

Best wished for you and your family!

Yours sincerely,

Corresponding author : Hanyan Li

E-mail address: liyany_2005@163.com

Reviewer 2 Report

This paper presents an interesting fabrication process for WR-2.2 waveguide. However, some questions appears as listed in the following.

1.     The full name for UV-LIGA is not correct.

2.     The 7th step in Fig. 2 should be added. Also, number (1) – (7) should be marked on the figure.

3.     What is the meaning of OM in paper?

4.     It can be noticed that the fabricated waveguide is inserted into a CNC machined metal flange. It is obvious that the aperture on the flanges is slightly larger than the size of the UV-LIGA fabricated waveguide for safely insertion. There must exist a small gap between them. How to make sure the waveguide is stably assembled? Also, alignment to external waveguide devices is also a problem, assembly tolerance exists in the assembly of waveguide and flange. How is this problem solved?

5.     How is the accuracy in paper defined? Is this the largest fabrication tolerance?

6.     In each sub-section of section 4, it is better to add a table to summarize the steps in each procedure.

7.     The organization of the paper is not very good. Section 3 and 4 have many repeat contents. Authors should re-organize the logic of the paper and highlight the different between them.

8.     A scale ruler or reference object can be added in Fig. 11. Equations should be numbered.

9.     Authors mentioned that the roughness is about 79.6 nm. However, the largest magnitude shown in Fig. 12 is 523.2nm. Please explain this.

10.   How is the return loss performed? Reviewer suggests a re-simulation of the waveguide by considering the test error, dimensional tolerance and test error and comparison with the measured responses.

11.   A table should be added to compare this work with the reference works presenting in the introduction, e.g., Insertion loss, return loss, surface roughness, effective conductivity and so on.

Author Response

Dear Editor ,

        We have studied your comments carefully and made a careful revision on the original manuscript. We would like to submit a revised manuscript which has revised according to the your and reviewers’ comments, a detailed point-by-point reply to all of the reviewers' comments, and a marked up manuscript where we highlight in red the changes made with respect to the initial submission. We hope you can reconsider our revised manuscript for potential publication in Micromachines.

All authors have read and approved this version of the article, and due care has been taken to ensure the integrity of the work. Neither the entire paper nor any part of its content has been published or has been accepted elsewhere. It is not being submitted to any other journal.

Thank you very much for providing a complete guide to the manuscript revision and submission. We are looking forward to hearing from you soon.

Best wished for you and your family!

Yours sincerely,

Corresponding author : Hanyan Li

E-mail address: liyany_2005@163.com

Round 2

Reviewer 1 Report

I still have a several questions to the authors:

1. As the authors state in a response letter, the main advantage of using SU-8 based UV-LIGA technology compared to using KMPR is better stability and repeatability of results, especially for structures with the thickness up to 400 mkm. However, the article lacks the necessary data to support this. In Fig. 11, several samples of waveguides are shown, however, measurements for the surface roughness were carried out only for one of them, and only at one point. In the response letter said that “We have measured several areas in different waveguides. The results show the roughness is approximately the same.”

I guess the authors should include these measurements in the article to determine and refine the statistics, since this is an important result. At least one waveguide can be measured at different precisions to prove stability.

2. The introduction section needs to be expanded. Several papers when KMPR based UV-LIGA technology is employed (and smaller roughness is obtained) were cited in the introduction, but there are no comments and no comparison with the results in this paper.

3. Another question is related to the results of roughness testing in the Fig.12. What did the authors mean by "sidewall roughness is 79.6nm"? One of the axes has a value of 523nm, does this mean that there are separate roughnesses with higher magnitude, and the average value is 79.6nm? Is it possible to specify the variance? Or does the wall roughness not exceed 79.6 nm? The authors are asked to clarify the issue. 

4. Data from Fig. 15 and 16 should be combined on one graph in order to visually relate the calculations and the experiment.

Author Response

Dear Editor,

        We have studied your comments carefully and made a careful revision on the original manuscript. We would like to submit a revised manuscript which has revised according to the your and reviewers’ comments, a detailed point-by-point reply to all of the reviewers' comments, and a marked up manuscript where we highlight in red the changes made with respect to the initial submission. We hope you can reconsider our revised manuscript for potential publication in Micromachines.

All authors have read and approved this version of the article, and due care has been taken to ensure the integrity of the work. Neither the entire paper nor any part of its content has been published or has been accepted elsewhere. It is not being submitted to any other journal.

Thank you very much for providing a complete guide to the manuscript revision and submission. We are looking forward to hearing from you soon.

Best wished for you and your family!

Yours sincerely,

Corresponding author : Hanyan Li

E-mail address: liyany_2005@163.com

Reviewer 2 Report

Firstly, I don’t think it is appropriate to use annotation on previous version of the manuscript. The authors should update a new version and highlight their actions in their new submission. It makes the manuscript unclear especially in Section 3.

Some responses to the questions are not satisfied.

Response 3: It is not a correct style to define an abbreviation. Please check.

Response 4: The author did not reply the last questions directly. Now we can notice that alignment tolerance exists between two parts. If the authors hold a view that waveguide channel is tightly fit to the flange part, more evidence should be presented. How can make sure the waveguide aperture positioning right at the center of the waveguide flange?

Response 5: The definition of the accuracy should be added in the paper.

Response 8: Reference ruler did not add in the figure. Please add it.

Response 9: I don’t know why authors provide some definitions. Can author present the return loss namely S11 response of the waveguide?

Since the author has obtained the measured dimension tolerances and surface roughness, it is should not difficult to use those values in their simulation. Please provide re-simulated waveguide responses comparing with the measured ones.

Response 11: The response did not answer the question. A table should be added. In the table, the performance of the fabrication waveguide in this paper compares with the waveguides’ performances (return loss, insertion loss, surface roughness, effective conductivity, etc.) presenting in the reference works, such as reference 9-18.

Author Response

(The authors gave the same response as above.)

Round 3

Reviewer 1 Report

I recommend to accept the article. 
